# Multicentre randomised controlled trial: protocol for Plasma-Lyte Usage and Assessment of Kidney Transplant Outcomes in Children (PLUTO)

Wesley Hayes [1,2] Emma Laing [3] Claire Foley,[3] Laura Pankhurst,[3] Helen Thomas,[3] Helen Hume-Smith,[2] Stephen Marks,[1,2] Nicos Kessaris [4] William A Bryant,[5] Anastassia Spiridou,[5] Jo Wray [6] Mark J Peters[2,7]

**Correspondence to**
Dr Wesley Hayes;
Wesley.hayes@gosh.nhs.uk

## ABSTRACT

**Introduction** Acute electrolyte and acid–base imbalance is experienced by many children following kidney transplantation. When severe, this can lead to complications including seizures, cerebral oedema and death. Relatively large volumes of intravenous fluid are administered to children perioperatively in order to establish perfusion to the donor kidney, the majority of which are from living and deceased adult donors. Hypotonic intravenous fluid is commonly used in the post-transplant period due to clinicians' concerns about the sodium, chloride and potassium content of isotonic alternatives when administered in large volumes. Plasma-Lyte 148 is an isotonic, balanced intravenous fluid that contains sodium, chloride, potassium and magnesium with concentrations equivalent to those of plasma. There is a physiological basis to expect that Plasma-Lyte 148 will reduce the incidence of clinically significant electrolyte and acid–base abnormalities in children following kidney transplantation compared with current practice.

The aim of the PLUTO trial is to determine whether the incidence of clinically significantly abnormal plasma electrolyte levels in paediatric kidney transplant recipients will be different with the use of Plasma-Lyte-148 compared to intravenous fluid currently administered.

**Methods and analysis** PLUTO is a pragmatic, open-label, randomised controlled trial comparing Plasma-Lyte 148 to current care in paediatric kidney transplant recipients, conducted in nine UK paediatric kidney transplant centres. A total of 144 children receiving kidney transplants will be randomised to receive either Plasma-Lyte 148 (the intervention) intraoperatively and postoperatively, or current fluid. Apart from intravenous fluid composition, all participants will receive standard clinical transplant care. The primary outcome measure is acute hyponatraemia in the first 72 hours post-transplant, defined as laboratory plasma sodium concentration of <135 mmol/L. Secondary outcomes include symptoms of acute hyponatraemia, other electrolyte and acid–base imbalances and transplant kidney function.

The primary outcome will be analysed using a logistic regression model adjusting for donor type (living vs deceased donor), patient weight (<20 kg vs ≥20 kg pretransplant) and transplant centre as a random effect.

### Strengths and limitations of this study

► This is the first randomised controlled trial to evaluate balanced isotonic intravenous fluid in paediatric kidney transplant recipients.
► A process evaluation will be used to tailor the trial to optimise children and families' experience of participating.
► Variation in current clinical practice between centres and clinicians makes standardisation of fluid administration in the control arm impractical, which could potentially affect the effect size.
► Blinding was not considered necessary due to the objective nature of outcome measures.

**Ethics and dissemination** The trial received Health Research Authority approval on 20 January 2020. Findings will be presented to academic groups via national and international conferences and peer-reviewed journals. The patient and public involvement group will play an important part in disseminating the study findings to the public domain.
**Trial registration numbers** 2019-003025-22 and 16586164.

## INTRODUCTION

Kidney transplantation is the treatment of choice for children with end-stage kidney disease.[1] Currently, acute electrolyte and acid–base imbalance is experienced by many paediatric recipients in the post-operative period.[2] Abnormalities include acute hyponatraemia, hyperkalaemia and metabolic acidosis; smaller recipients of deceased donor kidneys are most at risk.[2]

Acute hyponatraemia is closely associated with clinical complications in children after transplant: cerebral oedema, seizures[3] and death.[4] Up to 8% of children suffer hyponatraemia-related seizures following kidney transplant.[3] Pre-pubertal children are

at particular risk of complications from acute hyponatraemia because of the relatively low ratio of cerebrospinal fluid and brain volume compared with older children and adults.[5 6] Acute hyponatraemia also predicts symptoms such as nausea and vomiting, headaches, breathlessness and reduced conscious level.[7 8]

Several factors predispose children to acute hyponatraemia following kidney transplant:

First, relatively large volumes of intravenous fluid are administered perioperatively in order to establish and maintain perfusion to the adult donor kidney.[9] Second, hypotonic intravenous fluid is predominantly administered postoperatively in paediatric kidney transplant centres.[2] Thirdly, the post-operative period is characterised by an acute antidiuretic response which limits water excretion.[8 10]

Children receiving isotonic fluid as opposed to hypotonic fluid for indications other than kidney transplant experience less hyponatraemia.[11–15] A randomised trial in children with sepsis comparing balanced fluid to 0.9% sodium chloride is currently underway (Pragmatic Paediatric Trial of Balanced vs Normal Saline Fluid in Sepsis (PRoMPT BOLUS) study, Clinicaltrials.gov NCT03340805).[16] National guidance now recommends isotonic fluid for children in the general paediatric setting.[17] However, this guidance has not been adopted in paediatric transplant care because of a view that the period following kidney transplantation differs from other situations in terms of the volume of fluid administered and the associated sodium and chloride load. Hence, hypotonic fluid is still routinely administered to children following kidney transplantation[2 3] and the use of isotonic fluid in this context requires evaluation.

Plasma-Lyte-148 is an isotonic, balanced intravenous fluid.[18] It is licenced in children and contains sodium, chloride, potassium, magnesium with concentrations equivalent to those of plasma. There is a physiological basis to expect that Plasma-Lyte-148 will reduce the incidence of clinically significant electrolyte and acid–base abnormalities in children following kidney transplantation compared with current standard fluid.

A perceived risk of hyperkalaemia from the potassium content of Plasma-Lyte concerns some paediatric nephrologists, although a systematic review in adult kidney transplant recipients demonstrated less hyperkalaemia and acidosis with Plasma-Lyte compared with 0.9% sodium chloride.[19] A randomised trial[20] and a retrospective analysis in adult kidney transplant recipients[21] found a lower risk of hyperkalaemia with Plasma-Lyte-148 compared with 0.9% sodium chloride.

Currently, there is equipoise regarding use of balanced isotonic fluid versus hypotonic and chloride rich solutions for children undergoing kidney transplant.

The aim of the Plasma-Lyte Usage and Assessment of Kidney Transplant Outcomes in Children (PLUTO) trial is to determine whether the incidence of abnormal and potentially dangerous plasma electrolyte levels in paediatric kidney transplant recipients will be different with

**Table 1** Participating paediatric kidney transplant centres

| Trust | Principal investigator |
| --- | --- |
| Royal Belfast Hospital for Sick Children | Dr Mairead Convery |
| Birmingham Childrens Hospital | Dr Mordi Mourah |
| Bristol Royal Hospital for Children | Dr Jan Dudley |
| Great Ormond Street Hospital | Dr Wesley Hayes |
| Evelina London Children's Hospital | Dr Nick Ware |
| Leeds Children's Hospital | Dr Pallavi Yadav |
| Royal Manchester Children's Hospital | Dr Mohan Shenoy |
| Great North Children's Hospital | Dr Michal Malina |
| Nottingham Children's Hospital | Dr Martin Christian |
| University Hospital of Wales* | Dr Shivaram Hegde |
| University Hospital Southampton* | Dr Shuman Haq |
| Alder Hey Children's Hospital* | Dr Henry Morgan |

*Participant identification centre.

the use of Plasma-Lyte-148 compared with current standard intravenous fluids.

## METHODS AND ANALYSIS
### Study design and setting
PLUTO is an open-label randomised controlled trial comparing Plasma-Lyte 148 (the intervention) to current intravenous fluids in paediatric kidney transplant recipients, conducted in nine UK paediatric kidney transplant centres plus three participant identification centres (table 1). The study started on 8 June 2020, with planned completion on 31 October 2022.

### Intervention and comparator
Plasma-Lyte 148 solution for infusion and Plasma-Lyte 148 and glucose 5% w/v solution for infusion are licensed in children and used in paediatric intensive care, surgical and general paediatric wards. They contain sodium 140 mmol/L, chloride 98 mmol/L, potassium 5 mmol/L, magnesium 1.5 mmol/L and gluconate/acetate buffer to maintain physiological pH (table 2).[18] Both solutions can be administered interchangeably within the intervention arm.

The comparator is the current paediatric kidney transplant intravenous fluid regimen; this varies by paediatric transplant centre and individual clinician. Many paediatric transplant centres use Hartmann's fluid or 0.9% sodium chloride intraoperatively, followed by a combination of 0.45% sodium chloride and 0.9% sodium chloride with varying concentrations of glucose postoperatively. Some centres use 4.5% human albumin solution in addition.[2] The variability in standard clinical practice has arisen from the lack of evidence to inform choice of intravenous fluid in this patient population. For this pragmatic trial, permitting variation in the control arm is necessary

**Table 2** Electrolyte composition of Plasma-Lyte 148 and commonly used standard care fluids

| | Na$^+$ (mmol/L) | K$^+$ (mmol/L) | Mg$^{2+}$ (mmol/L) | Cl$^-$ (mmol/L) | Acetate (mmol/L) | Gluconate (mmol/L) |
|---|---|---|---|---|---|---|
| Plasma-Lyte 148 | 140 | 5 | 1.5 | 98 | 27 | 23 |
| 0.9% saline | 154 | | | 154 | | |
| 0.45% saline 5% glucose | 77 | | | 77 | | |

to achieve a robust comparison of the intervention to current clinical care.

## Outcome measures

The primary outcome measure is acute hyponatraemia within the first 72 hours post-transplant, defined as laboratory plasma sodium concentration of <135 mmol/L confirmed on a concurrent blood gas sample where feasible.

Secondary outcome measures are as follows:

► Symptoms of acute hyponatraemia (nausea, vomiting, headache and seizures) within the first 72 hours post-transplant.
► The degree of fluid overload experienced (defined as maximal proportional acute increase in patient weight from pretransplant weight in the first 72 hours post-transplant).
► Time to discharge from hospital (measured from transplant operation start time ('knife to skin') in days).
► Transplant kidney function at 1, 3, 7 and 90 days (estimated glomerular filtration rate (eGFR) calculated using creatinine-based univariate Schwartz formula[22]).
► Other electrolyte abnormalities within the first 72 hours post-transplantation:
  – Hypernatraemia (defined as plasma sodium concentration >145 mmol/L).
  – Hyperkalaemia (defined as plasma potassium concentration >5.5 mmol/L).
  – Hypokalaemia (defined as plasma potassium concentration <3.5 mmol/L).
  – Non-anion-gap acidosis (defined as plasma bicarbonate <20 mmol/L and anion gap <20 mmol/L).
  – Hyperglycaemia (defined as random blood glucose >5.5 mmol/L).
  – Hypomagnesaemia (defined as plasma magnesium concentration <0.7 mmol/L).
  – Hyperchloraemia (defined as plasma chloride concentration >107 mmol/L).
  – Excessive rate of reduction in plasma sodium concentration (defined as >1 mmol/L/hour averaged over 6 hours).
  – Excessive magnitude of reduction in plasma sodium concentration (defined as >10 mmol/L from pretransplant level).
  – Maximum and minimum systolic blood pressure (normalised to age and height percentile)

sustained on three repeated values on each day, for 3 days post-transplant.
  – Number of changes in intravenous fluid composition within the first 72 hours post-transplant.

## Eligibility criteria
### Inclusion criteria
► Patients under 18 years of age at the time of transplantation with valid patient/parental consent.
► Patients receiving a kidney-only transplant from either a living or a deceased donor, in a participating UK centre.

### Exclusion criteria
► Multiorgan transplant recipients.

## Screening, recruitment and randomisation

All patients <18 years old awaiting kidney transplantation, from either a deceased or living donor, will be screened for participation. The trial will be discussed during the process of preparation for kidney transplant in order to allow adequate time for families to consider participation. The potential benefits and risks of study participation are included in the study information sheets.

In accordance with Clinical Trial Regulations, patients aged under 16 years are deemed to be 'minors' and an appropriate adult must give consent on behalf of the child. The wishes of the child must form part of the decision-making process and the child's assent will be sought. For young people aged 16–18 years, their informed consent will be taken.

Participants will be randomised 1:1 to the intervention or control groups via an online system.[23] Randomisation will be stratified by transplant centre and patient weight (<20 kg vs ≥20 kg) and will further be balanced within blocks of varying, undisclosed sizes. The randomisation list will be produced by the trial statistician using SAS statistical software version 9.4M5.

## Treatment

Participants randomised to the intervention arm will receive either Plasma-Lyte 148 or Plasma-Lyte 148 and glucose 5%, which can be administered interchangeably at the discretion of the clinical team, both intraoperatively and postoperatively (figure 1). Participants randomised to the control arm will receive the standard intravenous fluids used at the site. Neither Plasma-Lyte 148 nor Plasma-Lyte 148 and glucose 5% should be administered

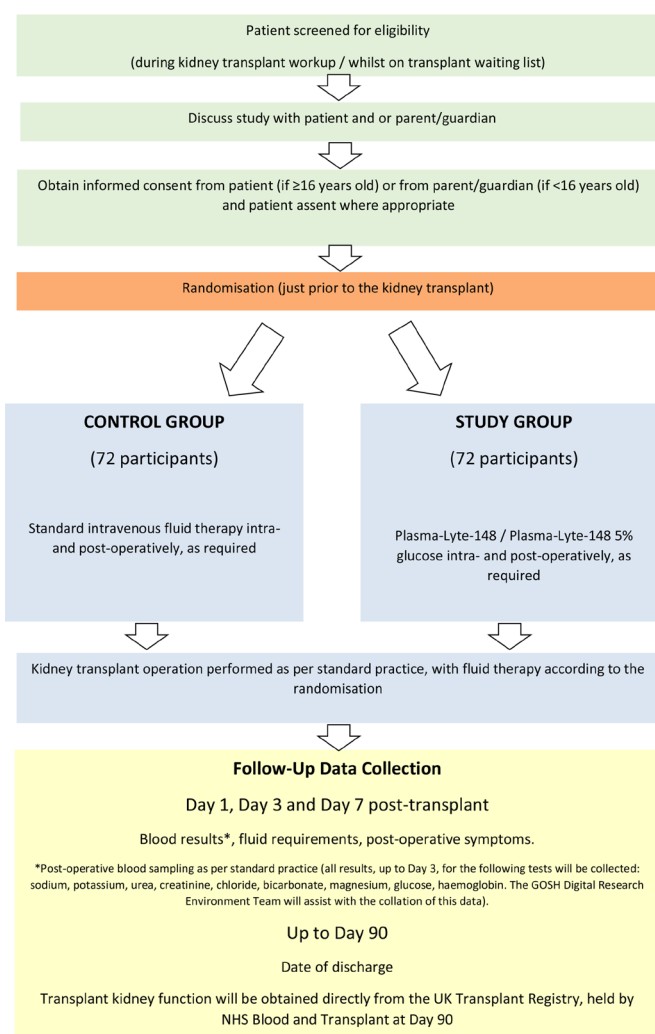

**Figure 1** Plasma-Lyte Usage and Assessment of Kidney Transplant Outcomes in Children trial flowchart.

in the control arm. The volume and rate of intravenous fluid (control or intervention) will be determined at the discretion of the treating clinician. The decision to discontinue intravenous fluid and switch to enteral fluid will be made by the treating clinician.

Apart from intravenous fluid composition, all participants will receive routine clinical transplant care as summarised in the schedule of procedures (table 3).

### Recruitment process evaluation (qualitative substudy)

A process evaluation will explore patients' and families' perceptions of the participant information, recruitment and consent process and acceptability of the trial.

At the time of initial approach for participation in PLUTO, families will be given a questionnaire (figure 2A), completion of which will be voluntary. A follow-up questionnaire will be given to families who provide initial consent for the PLUTO trial at days 4–7 post-transplant (figure 2B).

Both questionnaires include an option to opt-in to being contacted for a telephone interview. Interviews will explore experience of recruitment and consent,

participant information, motivation for participation (or reasons for not participating) and reasons for changes in decision making.

### Data collection
#### Baseline assessments
Baseline assessments comprise those undertaken as part of routine clinical care prior to kidney transplant:

► Eligibility checklist.
► Baseline form (pretransplant): details of the end stage-kidney disease (cause, dialysis type, dialysis start date and native urine output), actual patient weight (kilogram), patient height (centimetre) and blood pressure.

Baseline demographic data (patient age, sex and ethnicity) will be obtained directly from the UK Transplant Registry (UKTR), held by NHS Blood and Transplant (NHSBT).

#### On the day of transplant
► Randomisation form (on day of transplant): randomisation date/time, randomisation ID number, treatment arm.
► Pretransplant blood test results (table 4).

#### Post-transplant (72 hours)
The 72 hour post-transplant period starts at the time of transplant operation 'knife to skin'.
► Transplant operation form: operation start time, duration and graft placement.
► Fluids: all fluid input/output, and all patient weight measurements in the first 72 hours post-transplant
► Blood products: all blood products administered in the first 72 hours post-transplant.
► Medications: all inotropes, diuretics, immunosuppressives, insulin, antiemetics and electrolyte supplements administered in the first 72 hours post-transplant.
► Symptom assessment: daily symptom assessment for nausea, vomiting, headache, pain score and seizures. Pain is graded using an age-appropriate scale (Face, Legs, Activity, Cry, Consolability scale (FLACC) or Wong-Baker).[24 25]
► Blood pressure: maximum and minimum systolic blood pressure sustained on three repeated values on each day, for 3 days post-transplant.
► Blood test results: all results to 72 hours post-transplant will be collected (table 4).

#### Follow-up data collection
► Post-transplant form: need for dialysis post-transplant, creatinine at day 7 post-transplant (±1 day).
► Hospital discharge: date of discharge from hospital.

No additional outpatient visits will be required for the research study. Plasma creatinine concentration at month 3 (±14 days) will be obtained directly from the UKTR held by NHSBT.

**Table 3** Schedule of procedures

| | Screening | Baseline (Pre-Tx) | At Tx | Day 1 Post-Tx | Day 2 Post-Tx | Day 3 Post-Tx | Day 7 Post-Tx | Month 3 Post-Tx | Hospital discharge |
|---|---|---|---|---|---|---|---|---|---|
| **Enrolment** | | | | | | | | | |
| Qualitative substudy—give Q1 | X | | | | | | | | |
| Eligibility assessment | X | | | | | | | | |
| Informed consent | | X | | | | | | | |
| Baseline characteristics | | X | | | | | | | |
| Reconfirm consent/assent | | | X | | | | | | |
| Randomisation | | | X | | | | | | |
| Qualitative substudy—give Q2 | | | | | | | X | | |
| **Treatment** | | | | | | | | | |
| Administration of trial fluid | | | X | X | X | X | | | |
| **Assessments** | | | | | | | | | |
| Transplant operation data | | | X | | | | | | |
| Blood results† | | X | All results for 72 hours from transplant 'knife-to-skin' | | | | | | |
| Blood gas results† | | X | All results for 72 hours from transplant knife-to-skin | | | | | | |
| Patient weight | | X | All weights for 72 hours from transplant knife-to-skin | | | | | | |
| Fluid data | | | All fluid for 72 hours from transplant knife-to-skin | | | | | | |
| Medication data | | | All meds* for 72 hours from transplant 'knife-to-skin' | | | | | | |
| Symptom assessment | | | | X | X | X | | | |
| Dialysis | | | | X | X | X | | | |
| Blood pressure | X | | | X | X | X | | | |
| Safety reporting | | | X | X | X | X | | | |
| Transplant graft function | | | | | | | X | X‡ | |
| Discharge date | | | | | | | | | X |

No additional tests are required for Plasma-Lyte Usage and Assessment of Kidney Transplant Outcomes in Children trial participants. This table summarises the data required at each timepoint.

*Data must be collected on all inotropes, diuretics, immunosuppressives, insulin, antiemetics and electrolyte supplements administered during this 72-hour period.

†Blood results and Venus blood gas data will be obtained directly from the Trust Pathology System (*where possible*), anonymised and then transferred to the Digital Research Environment at Great Ormond Street Hospital.

‡Data will be obtained directly from the UK Transplant Registry, held by NHS Blood and Transplant (eg, patient ethnicity, blood group, donor details, matching details and transplant graft function).

## Statistical plan
### Sample size
The sample size required is 144 participants randomised.

The baseline incidence was determined by analysis of 76 paediatric transplants performed at Great Ormond Street Hospital between 1 January 2015 and 31 March

QUESTIONNAIRE 1 – BEFORE THE TRANSPLANT

| | Strongly agree | Agree | Neither agree nor disagree | Disagree | Strongly disagree |
|---|---|---|---|---|---|
| I was satisfied with how I was **approached** about PLUTO | ○ | ○ | ○ | ○ | ○ |
| I understood the **information** that I received from the doctor/research nurse about PLUTO | ○ | ○ | ○ | ○ | ○ |
| I had sufficient opportunity to ask **questions** about PLUTO | ○ | ○ | ○ | ○ | ○ |
| Any questions I had about PLUTO were answered in a way that I could **understand** | ○ | ○ | ○ | ○ | ○ |
| I was satisfied with the **consent** process for PLUTO | ○ | ○ | ○ | ○ | ○ |
| I had enough **time** to think about whether or not to consent to my child participating in PLUTO | ○ | ○ | ○ | ○ | ○ |

QUESTIONNAIRE 2 – AFTER THE TRANSPLANT

| | Strongly agree | Agree | Neither agree nor disagree | Disagree | Strongly disagree |
|---|---|---|---|---|---|
| I was satisfied with how I was **approached** about PLUTO on the night of transplant | ○ | ○ | ○ | ○ | ○ |
| I remembered the **information** I had previously been given about PLUTO | ○ | ○ | ○ | ○ | ○ |
| I had sufficient opportunity to ask **questions** about PLUTO on the night of transplant | ○ | ○ | ○ | ○ | ○ |
| Any questions I had about PLUTO were answered in a way that I could **understand** | ○ | ○ | ○ | ○ | ○ |
| I was satisfied with the **re-consent** process for PLUTO on the night of transplant | ○ | ○ | ○ | ○ | ○ |
| I was happy with my original **decision** to consent to my child participating in PLUTO | ○ | ○ | ○ | ○ | ○ |

**Figure 2** Qualitative substudy questionnaires. PLUTO, Plasma-Lyte Usage and Assessment of Kidney Transplant Outcomes in Children.

**Table 4** Blood parameters

| Test | Preferred units |
|---|---|
| Plasma sodium level | mmol/L |
| Plasma potassium level | mmol/L |
| Plasma urea level | mmol/L |
| Plasma creatinine level | micromol/L |
| Plasma chloride level | mmol/L |
| Plasma bicarbonate level | mmol/L |
| Plasma magnesium level | mmol/L |
| Plasma calcium level | mmol/L |
| Plasma inorganic phosphate level | mmol/L |
| Blood glucose level | mmol/L |
| Haemoglobin mass concentration in blood | g/L |
| Blood gas pH | – |
| Blood gas–bicarbonate | mmol/L |
| Blood gas–chloride | mmol/L |
| Blood gas–sodium | mmol/L |
| Blood gas–potassium | mmol/L |
| Blood gas–ionised calcium | mmol/L |
| Blood gas–glucose | mmol/L |

test for harm or benefit will be conducted after 70 and 100 participants have been recruited into the trial with primary outcome data. The stopping rules will be used as a guideline, alongside the other safety data available to the data monitoring committee (DMC), and used as part of their overall assessment of the trial.

## Analysis
The analyses will be described in detail in a full statistical analysis plan. This section summarises the main issues.

The population used for efficacy analyses will be a modified intention-to-treat analysis which will include all randomised participants who receive a transplant, as although unlikely it would be illogical to include any participant who did not receive a transplant after being randomised. This will be the primary analysis for the trial, though the primary outcome will also be analysed per protocol.

Characteristics of all randomised patients will be tabulated by arm of the trial to describe the cohort. A Consolidated Standards of Reporting Trials diagram will be presented to show the flow of patients through the trial.

## Primary outcome
This includes the proportion of patients who experience hyponatraemia (defined as plasma sodium concentration <135 mmol/L) within 72 hours post-transplant. This will be analysed using a logistic regression model adjusting for donor type (living vs deceased donor), patient weight (<20 kg vs ≥20 kg pretransplant) and transplant centre as a random effect. Subgroup and sensitivity analyses will

2018, which showed that 45 children (59%) experienced hyponatraemia within the first 72 hours post-transplant when managed with 0.45% saline 5% glucose fluid.[2] The effect size was estimated from the Cochrane systematic review comparing isotonic to hypotonic maintenance fluid in children, in which a 50% reduction in risk of hyponatraemia was shown.[11] A greater effect size is anticipated because the rate of fluid delivery to children following kidney transplant significantly exceeds maintenance requirements.[9] Using the 50% effect size from the Cochrane review, we found that the assumed incidence of hyponatraemia would be reduced to 29.5%. A two-sided test with 90% power, 5% type I error and 1:1 allocation, and allowing for two formal interim analyses for harm or benefit would require 128 participants. After allowing for 10% attrition, for example, if a transplant cannot proceed due to a positive cross match, the total number of participants required is 144.

## Interim analyses
A group sequential design with O'Brien-Fleming boundaries will inform early stopping of the trial in the case of strong evidence of harm or benefit, while preserving the overall 5% type I error rate. Two interim analyses to

be used to assess whether the treatment effect differs according to the fluid regimen used in the standard arm.

## Secondary outcomes

The secondary outcomes which assess the incidence of other electrolyte and acid–base disturbance within 72 hours post transplantation (hyperkalaemia, hypokalaemia, hypernatraemia, acidosis, hypomagnesaemia, hyperglycaemia and excessive rate and magnitude of fall in plasma sodium concentration) will all be analysed using the same model as the primary outcome. Exact logistic regression will be used to compare instances of nausea/vomiting, headache and seizures. Adjusted normal linear regression models will be used to assess differences in the continuous secondary outcomes, and an adjusted Cox proportional hazards model will be used to assess differences in time to discharge. The number of changes in intravenous fluid composition within the first 72 hours post-transplant will be summarised by treatment arm.

Any missing primary and secondary outcome data will be summarised. Primary and secondary outcome measures will not be imputed, and these will be treated as missing data and excluded from the relevant analyses. If outcome data are missing for more than 25% of participants, outcomes will not be reported. To explore if missing values have an undue impact on the primary outcome result, a sensitivity analysis using multiple imputation will be performed if the primary outcome is missing in more than 5% of the participants included in the modified intention-to-treat analysis.

## Patient and public involvement (PPI)

A kidney transplant patient and her mother are coapplicants for the grant and members of the Trial Steering Committee (TSC). They make sure that the trial remains patient-centred throughout its delivery and will lead the dissemination of results to patients and families. The research plan was also shaped by a workshop with the Young Persons' Advisory Group at Great Ormond Street Hospital, and involvement of five families with children awaiting kidney transplantation who advised on the burden of intervention.

## Ethics, safety and dissemination
### Ethical compliance

The investigator will ensure that this trial is conducted in accordance with the principles of the Declaration of Helsinki, ICH Good Clinical Practice Guidelines and in accordance with the terms and conditions of the ethical approval given to the trial. A favourable opinion has been given by a research ethics committee (HRA reference 19/LO/1866).

## Safety
### Adverse events and expected adverse drug reactions

Non-serious adverse events (SAEs) will not be reported. Plasma-Lyte 148 is a licensed product and therefore adverse reactions which are expected (assessed against the Summary of Product Characteristics[18]) and not serious do not require recording.

SAEs will be notified within 24 hours and monitored regularly by the DMC to ensure ongoing safety of trial participants.

The SAEs listed below are anticipated in this patient group, and have therefore been excluded from trial safety reporting:

► Surgical complications.
  – Return to theatre for exploration.
  – Haemorrhage.
  – Blood transfusion.
  – Transplant obstruction/dilatation/hydronephrosis.
  – Thrombosis (thrombosis of transplant, Deep Vein Thrombosis (DVT), pulmonary embolus and arteriovenous fistula).
  – Transplant kidney arteriovenous fistula.
  – Transplant nephrectomy.
  – Haematuria.
  – Cystoscopy for evacuation of haematoma or stent removal.
  – Renal artery stenosis.
  – Bowel injury, obstruction or ischaemia.
  – Lymphocele/collection±drainage.
  – Hernia.
  – Urine leak.
  – Nephrostomy.
► Infective complications.
  – Viraemia (Epstein Barr Virus (EBV), cytomegalovirus (CMV), JC and BK viraemia).
  – Bacteraemia.
  – Fever.
► Immunological complications.
  – Acute rejection.
  – Transplant kidney biopsy.
► Medication-related complications.
  – Calcineurin inhibitor nephrotoxicity and neurotoxicity.
  – New-onset diabetes.
► Other.
  – Delayed graft function/primary non-function.
  – Thrombocytopenia/thrombotic microangiopathy.

## Trial oversight

The Trial Management Group (TMG) will be responsible for the day-to-day running and management of the trial.

The TSC will provide overall supervision for the trial and provide advice through its independent chair. The ultimate decision on continuation of the trial lies with the TSC.

The DMC will be responsible for safety monitoring. Recommendations made by the DMC to alter the conduct of the study will be forwarded to the TSC for a final decision. The sponsor will forward such decisions to regulatory authorities, as appropriate. The DMC will be independent of the study team and will have no direct involvement in other aspects of the trial.

## Dissemination

The final study dataset will be analysed, and results will be published as soon as possible following completion of study follow-up, final data checks and database lock. The TMG will form the basis of the writing committee and will advise on the nature of publications.

Study findings will be presented to academic and non-academic groups. The PPI group will play an important part in disseminating the study findings into the public domain. Dissemination to non-academic audiences including service users, commissioners, clinicians and service providers will be facilitated through the use of existing networks, for example, email lists and social media.

All research teams and PPI members involved in the study will be invited to a close-out meeting to discuss the findings of the study.

Open-access, peer-reviewed academic outputs and research reports, together with associated summaries and key findings, will be produced for funders, policy makers and NHS audiences and held on the study website.

Any publications arising from this study will adhere to the National Institute for Health Research funding and support outputs guidance.

**Author affiliations**
[1]Department of Nephrology, Great Ormond Street Hospital for Children NHS Foundation Trust, London, UK
[2]Institute of Child Health, University College London, London, UK
[3]Clinical Trials Unit, NHS Blood and Transplant, Cambridge, UK
[4]Department of Nephrology and Transplantation, Guy's Hospital, London, UK
[5]Digital Research, Informatics and Virtual Environments Unit, Great Ormond Street Hospital for Children NHS Foundation Trust, London, UK
[6]Department of Health Psychology, Great Ormond Street Hospital for Children NHS Foundation Trust, London, UK
[7]Paediatric Intensive Care Unit, Great Ormond Street Hospital For Children NHS Trust, London, UK

**Correction notice** This article has been corrected since it was published Online First. The abstract section has been updated.

**Acknowledgements** This study is sponsored by Great Ormond Street Hospital for Children NHS Foundation Trust. The study was supported by the Digital Research Environment Team at Great Ormond Street Hospital. Our thanks to Maya and Sylwia Olejnik-Antkowiak, patient and parent representatives who have contributed to the design of this study, and to the NIHR Great Ormond Street Hospital Young Person's Advisory Group for research for codesigning patient information resources.

**Contributors** WH developed the idea for this study, drafted this manuscript and has oversight of the study with mentorship from MJP. WH, EL, LP, HT and MJP applied for funding and developed the trial protocol. JW developed the recruitment process evaluation. NK, HH-S, WAB, AS and SM are all study coinvestigators and contributed to the writing of this protocol. EL, CF and LP are the trial manager, clinical operations manager and trial statistician, respectively, at the NHS Blood and Transplant CTU and contributed to the writing of the protocol and study planning and approval process. All authors read and approved the final manuscript.

**Funding** This study is funded by the National Institute for Health Research (NIHR) Research for Patient Benefit programme (project reference number NIHR200512). The views expressed are those of the author(s) and not necessarily those of the NIHR or the Department of Health and Social Care.

**Competing interests** None declared.

**Patient and public involvement** Patients and/or the public were involved in the design, conduct, reporting or dissemination plans of this research. Refer to the Methods and analysis section for further details.

**Patient consent for publication** Not applicable.

**Provenance and peer review** Not commissioned; externally peer reviewed.

**Data availability statement** Data are available upon reasonable request. No data generated or analysed to date; this manuscript is the clinical trial protocol only. On completion of the trial, the fully anonymised dataset will be available upon request.

**ORCID iDs**
Wesley Hayes http://orcid.org/0000-0002-8091-6945
Emma Laing http://orcid.org/0000-0002-8309-0990
Nicos Kessaris http://orcid.org/0000-0002-4067-7003
Jo Wray http://orcid.org/0000-0002-4769-1211

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
