## [Reviewer comments · BMJ Open]

ARTICLE DETAILS

TITLE (PROVISIONAL)	Protocol for multi-centre randomised controlled trial: Plasma-Lyte Usage and assessment of kidney Transplant Outcomes in children (PLUTO)
AUTHORS	Hayes, Wesley; Laing, Emma; Foley, Claire; Pankhurst, Laura; Thomas, Helen; Hume-Smith, Helen; Marks, Dr. Stephen; Kessarlis, Nicos; Bryant, William A.; Spiridou, Anastassia; Wray, Jo; Peters, Mark

VERSION 1 – REVIEW

REVIEWER	Kanaris, Constantinos Cambridge University Hospitals NHS Foundation Trust
REVIEW RETURNED	05-Sep-2021

GENERAL COMMENTS	Thank you for inviting me to review the protocol to the PLUTO trial, this is a trial I have been looking forward to for some time and that has been much needed for some time in the paediatric population. I have no hesitation in recommending this for publication in BMJ Paediatrics pending some (very) minor corrections, it is a good fit for the journal. a) Obviously, the study protocol cannot be changed. The omission of measuring hyperchloraemia as a secondary outcome is a glaring one. If this could be included it would make the study even stronger as the association of hyperchloraemia with mortality and prolonged hospital stay, especially in critically ill children is widely reported. If this cannot be included then a clarification as to why it is not would be necessary. b) Furthermore, this is labelled as a “pragmatic” study, perhaps done so to encompass and acknowledge that there are different fluid protocols for different paediatric renal centres in each of the participating hospitals. Although this is in part addressed in the study limitations further transparency is needed on the following point. The primary endpoint is that of acute hyponatremia in the first 3 days following renal transplant. There is already, reportedly, variation between centres in terms of post-transplant hyponatraemia. Centres that use 0.45% saline as their maintenance fluid will have a higher incidence than those centres using 0.9% saline post-op. This point is important and needs to be made more transparent in the limitations section. c) The introduction makes reference to adult studies comparing balanced solutions VS 0.9% saline to renal transplant recipients. There are only a handful of these in the literature and some are not referenced, the following should also ideally be included. • O'Malley CM, Frumento RJ, Hardy MA, et al. A randomized, double-blind comparison of lactated Ringer's solution and 0.9%
--

	NaCl during renal transplantation. Anesth Analg. 2005;100(5):1518-24  • Khajavi MR, Etezadi F, Moharari RS, et al. Effects of normal saline vs. lactated ringer's during renal transplantation. Ren Fail. 2008;30(5):535-9 • Modi MP, Vora KS, Parikh GP, et al. A comparative study of impact of infusion of Ringer's Lactate solution versus normal saline on acid-base balance and serum electrolytes during live related renal transplantation. Saudi J Kidney Dis Transpl. 2012;23(1):135-7 d) The introduction also, appropriately, makes reference to the recent change of the Resus council guidance in paediatric shock to use balanced solution rather than saline as a first-choice fluid bolus. A further reference to the recent RCPCH review paper on the management of paediatric hyperkalaemia recommending balanced solutions for these patients should ideally also be included  • Rubens, M. and Kanaris, C., 2021. Fifteen-minute consultation: Emergency management of children presenting with hyperkalaemia. Archives of Disease in Childhood-Education and Practice. e) There is also reference to the different anion/cation content of each type of fluid. This is important and necessary. It would be much easier for the reader to conceptualise if these were also presented in table form. f) There is no section on consent, how consent will be obtained including what risks will be listed should be included. g) The study is ongoing, there are no dates included in the manuscript to clarify when the study started and when it is estimated to finish. Overall, the paper is written to a high standard, the data collection/sample size/safety and statistical aspects of it are scientifically robust Pending the minor revisions and clarifications suggested, I would very much recommend the protocol for publication.
--	---

REVIEWER	Ballard, Heather Ann and Robert H Lurie Children's Hospital of Chicago
REVIEW RETURNED	06-Sep-2021

GENERAL COMMENTS	This paper describes a protocol to determine whether Plasma-Lyte use versus standard of care causes less acute hyponatremia if administered during the perioperative period surrounding pediatric kidney transplant. Research question: The research question is clearly defined, "Will Plasma-Lyte 148 administration perioperatively result in less hyponatremia when compared to the standard of care?" Abstract: Well organized description of introduction, methods, and limitations. Ethics information is detailed within abstract. Introduction: Concise review of what is known about hyponatremic complications in children after renal transplant. Danger of using hypotonic solutions is well described in introduction; there is limited information reported about the dangers of using normal saline, except for potential hyperkalemia. Consider adding text regarding the acidemia caused by giving hyperchloremic concentration of normal saline. Also, consider detailing the secondary outcomes in the last paragraph of the introduction.
--

	Methods: Study design: Non-blinded randomized trial conducted at 9 different UK centers comparing Plasma-Lyte 148 to standard of care. Each center can control what the standard of care is for fluid management. The study design is reasonable but consider reporting results which fluids were utilized at each center. The plan is to randomize with an online system using stratification by transplant center and patient weight (<>20 kg). Sample size calculation of 144 is reasonable, accounting for 50% reduction in hyponatremia with an incidence of 59%. The 50% reduction in hyponatremia occurred when comparing hypotonic versus isotonic maintenance infusions. Since each center can control whatever maintenance fluid is used, it is unknown whether some centers use isotonic fluids. If several centers are already using isotonic solutions for maintenance, the effect size would be much lower and a much higher sample size would be required. Description: Inclusion criteria (less than 18 years old, kidney transplant only) and exclusion criteria(multi-organ transplant) are well defined. Assessments during the first week after transplant are well defined and should be consistently obtained with both study arms. It is reasonable to exclude patients who did not receive a kidney transplant in this modified intention to treat analysis. Ethics: Approved by Research Ethics Committee. It appears that there will be adequate time for parents and patients to consider participation. Process evaluation will be used to explore parents/patients experience of recruitment and participation. Plan to use a CONSORT diagram to describe flow of patients in trial. Patient and parent are included as members of research team with input from families of patients waiting for kidney transplant for recruitment materials. Outcomes: Primary Outcome: acute hyponatremia within 72 hours post-transplant Secondary Outcomes: hyponatremia symptoms, fluid overload, time to discharge, transplant kidney function, other electrolyte abnormalities, changes in IV fluid composition. Statistical Analysis: The primary outcome (hyponatremia) is planned to be analyzed with logistic regression adjusting for donor type, patient weight, and transplant center as a random effect. Sensitivity analysis will be used to determine whether treatment effect varies based on fluid regimen. There is a reasonable plan to deal with missing data: exclude missing data, but use sensitivity analyses to determine if multiple imputation leads to different results if missing data greater than 5%.
--	--

VERSION 1 – AUTHOR RESPONSE

Reviewer: 1

Dr. Constantinos Kanaris, Cambridge University Hospitals NHS Foundation Trust, Queen Mary University of London Barts and The London School of Medicine and Dentistry

Thank you for your comments to improve the manuscript which we have addressed below:

Comments to the Author:

Thank you for inviting me to review the protocol to the PLUTO trial, this is a trial I have been looking forward to for some time and that has been much needed for some time in the paediatric population. I have no hesitation in recommending this for publication in BMJ Paediatrics pending some (very) minor corrections, it is a good fit for the journal.

***Comment from the Editor: We are aware that the reviewer has provided the incorrect journal name in their report, but we have left this in as we do not edit peer review reports.

a) Obviously, the study protocol cannot be changed. The omission of measuring hyperchloraemia as a secondary outcome is a glaring one. If this could be included it would make the study even stronger as the association of hyperchloraemia with mortality and prolonged hospital stay, especially in critically ill children is widely reported. If this cannot be included then a clarification as to why it is not would be necessary.

Hyperchloraemia is already listed as a secondary outcome (see section: Methods and Analysis, Outcomes)

b) Furthermore, this is labelled as a “pragmatic” study, perhaps done so to encompass and acknowledge that there are different fluid protocols for different paediatric renal centres in each of the participating hospitals. Although this is in part addressed in the study limitations further transparency is needed on the following point. The primary endpoint is that of acute hyponatremia in the first 3 days following renal transplant. There is already, reportedly, variation between centres in terms of post-transplant hyponatraemia. Centres that use 0.45% saline as their maintenance fluid will have a higher incidence than those centres using 0.9% saline post-op. This point is important and needs to be made more transparent in the limitations section.

We have included this point as a limitation

c) The introduction makes reference to adult studies comparing balanced solutions VS 0.9% saline to renal transplant recipients. There are only a handful of these in the literature and some are not referenced, the following should also ideally be included.

- O'Malley CM, Frumento RJ, Hardy MA, et al. A randomized, double-blind comparison of lactated Ringer's solution and 0.9% NaCl during renal transplantation. *Anesth Analg*. 2005;100(5):1518-24
- Khajavi MR, Etezadi F, Moharari RS, et al. Effects of normal saline vs. lactated ringer's during renal transplantation. *Ren Fail*. 2008;30(5):535-9
- Modi MP, Vora KS, Parikh GP, et al. A comparative study of impact of infusion of Ringer's Lactate solution versus normal saline on acid-base balance and serum electrolytes during live related renal transplantation. *Saudi J Kidney Dis Transpl*. 2012;23(1):135-7

In order to enhance clarity, we restricted references to high quality prospective clinical trials and systematic reviews, so chose to exclude the above references

d) The introduction also, appropriately, makes reference to the recent change of the Resus council guidance in paediatric shock to use balanced solution rather than saline as a first-choice fluid bolus. A

further reference to the recent RCPCH review paper on the management of paediatric hyperkalaemia recommending balanced solutions for these patients should ideally also be included

• Rubens, M. and Kanaris, C., 2021. Fifteen-minute consultation: Emergency management of children presenting with hyperkalaemia. Archives of Disease in Childhood-Education and Practice.

This review was published after the trial protocol was developed, and after the protocol manuscript was submitted to BMJ open for publication (submitted July 2021). For this reason, we do not think it appropriate to retrospectively revise the protocol introduction to include it.

e) There is also reference to the different anion/cation content of each type of fluid. This is important and necessary. It would be much easier for the reader to conceptualise if these were also presented in table form.

f) There is no section on consent, how consent will be obtained including what risks will be listed should be included.

We have included consent and assent forms as appendices, and the following explanatory text can be found in the section “screening, recruitment and randomisation”:

All patients <18 years old awaiting kidney transplantation, from either a deceased or living donor, will be screened for participation. The trial will be discussed during the process of preparation for kidney transplant in order to allow adequate time for families to consider participation. The potential benefits and risks of study participation are included in the study information sheets

In accordance with Clinical Trial Regulations, patients aged under 16 years are deemed to be ‘minors’ and an appropriate adult must give consent on behalf of the child. The wishes of the child must form part of the decision-making process and the child’s assent will be sought. For young people aged 16 – 18 years, their informed consent will be taken.

g) The study is ongoing, there are no dates included in the manuscript to clarify when the study started and when it is estimated to finish.

We have added start and projected finish dates (Methods: The study started on 8 June 2020, with planned completion on 31 Oct 2022).

Overall, the paper is written to a high standard, the data collection/sample size/safety and statistical aspects of it are scientifically robust

Pending the minor revisions and clarifications suggested, I would very much recommend the protocol for publication.

Many thanks

Reviewer: 2

Dr. Heather Ballard, Ann and Robert H Lurie Children’s Hospital of Chicago

Many thanks for your detailed review and helpful feedback which we have addressed as follows:

Comments to the Author:

This paper describes a protocol to determine whether Plasma-Lyte use versus standard of care causes less acute hyponatremia if administered during the perioperative period surrounding pediatric kidney transplant.

Research question: The research question is clearly defined, “Will Plasma-Lyte 148 administration perioperatively result in less hyponatremia when compared to the standard of care?”

Abstract: Well organized description of introduction, methods, and limitations. Ethics information is detailed within abstract.

Introduction: Concise review of what is known about hyponatremic complications in children after renal transplant. Danger of using hypotonic solutions is well described in introduction; there is limited information reported about the dangers of using normal saline, except for potential hyperkalemia. Consider adding text regarding the acidemia caused by giving hyperchloremic concentration of normal saline. Also, consider detailing the secondary outcomes in the last paragraph of the introduction.

We agree that acidemia is an important complication of 0.9% sodium chloride, and have referenced acidaemia as a common complication in children receiving kidney transplants (introduction, paragraph 1) and have added text referencing increased risk of acidemia in RCTs in adult kidney transplant recipients (introduction, paragraph 7).

Methods:

Study design: Non-blinded randomized trial conducted at 9 different UK centers comparing Plasma-Lyte 148 to standard of care. Each center can control what the standard of care is for fluid management. The study design is reasonable but consider reporting results which fluids were utilized at each center. The plan is to randomize with an online system using stratification by transplant center and patient weight (<20 kg).

Each centre uses a combination of fluids in the comparator arm, and this can vary from patient to patient. We are collecting data on fluids administered and will report this with the trial results. In this protocol manuscript, it is not feasible to report fluid use in the comparator arm because practice is not standardised by centre.

Sample size calculation of 144 is reasonable, accounting for 50% reduction in hyponatremia with an incidence of 59%. The 50% reduction in hyponatremia occurred when comparing hypotonic versus isotonic maintenance infusions. Since each center can control whatever maintenance fluid is used, it is unknown whether some centers use isotonic fluids. If several centers are already using isotonic solutions for maintenance, the effect size would be much lower and a much higher sample size would be required.

We are aware of this and it was considered in detail before the study began. We undertook a survey of fluid used at participating centres when planning the trial. Although various combinations of fluids are used, most centres stated that they use predominantly hypotonic fluid in the postoperative period. In addition, larger relative fluid volumes are administered to children post kidney transplant than in the general paediatric setting in which a 50% reduction in hyponatraemia was observed, hence a greater effect size if anticipated in this trial. For these reasons, we felt that this sample size calculation is reasonable. In addition, we plan to undertake a subgroup analysis of the primary outcome to assess whether the treatment effect differs according to the fluid regimen used in the standard arm.

Description:

Inclusion criteria (less than 18 years old, kidney transplant only) and exclusion criteria (multi-organ transplant) are well defined. Assessments during the first week after transplant are well defined and should be consistently obtained with both study arms. It is reasonable to exclude patients who did not receive a kidney transplant in this modified intention to treat analysis.

Ethics: Approved by Research Ethics Committee. It appears that there will be adequate time for parents and patients to consider participation. Process evaluation will be used to explore

parents/patients experience of recruitment and participation. Plan to use a CONSORT diagram to describe flow of patients in trial. Patient and parent are included as members of research team with input from families of patients waiting for kidney transplant for recruitment materials.

Outcomes:

Primary Outcome: acute hyponatremia within 72 hours post-transplant

Secondary Outcomes: hyponatremia symptoms, fluid overload, time to discharge, transplant kidney function, other electrolyte abnormalities, changes in IV fluid composition.

Statistical Analysis: The primary outcome (hyponatremia) is planned to be analyzed with logistic regression adjusting for donor type, patient weight, and transplant center as a random effect. Sensitivity analysis will be used to determine whether treatment effect varies based on fluid regimen. There is a reasonable plan to deal with missing data: exclude missing data, but use sensitivity analyses to determine if multiple imputation leads to different results if missing data greater than 5%.

VERSION 2 – REVIEW

REVIEWER	Kanaris, Constantinos Cambridge University Hospitals NHS Foundation Trust
REVIEW RETURNED	30-Dec-2021

GENERAL COMMENTS	The majority of recommended revisions have been addressed. These include references to meta-analyses comparing balanced vs saline solution in adult renal transplant recipients, an improved consent section and the dates of the ongoing study. Two points from the initial review have not been addressed, 1) The introduction also, appropriately, makes reference to the recent change of the Resus council guidance in paediatric shock to use balanced solution rather than saline as a first-choice fluid bolus. A further reference a review paper on the management of paediatric hyperkalaemia recommending balanced solutions for these patients should ideally also be included 2) There is also reference to the different anion/cation content of each type of fluid. This is important and necessary. It would be much easier for the reader to conceptualise if these were also presented in table form These need to be rectified prior to publication as they will make the script stronger and easier to follow.
--

VERSION 2 – AUTHOR RESPONSE

1) The introduction also, appropriately, makes reference to the recent change of the Resus council guidance in paediatric shock to use balanced solution rather than saline as a first-choice fluid bolus. A further reference a review paper on the management of paediatric hyperkalaemia recommending balanced solutions for these patients should ideally also be included.

This NIHR funded clinical trial protocol is designed to provide robust evidence as to whether electrolyte abnormalities such as acute hyperkalaemia may be less common with balanced solutions in this group of children. To our knowledge, there are currently no clinical trial data or meta-analyses to cite which justify balanced solutions for these paediatric patients.

2) There is also reference to the different anion/cation content of each type of fluid. This is important

and necessary. It would be much easier for the reader to conceptualise if these were also presented in table form

We have added a table with fluid data as suggested (table 2).